# Gluten-Free Sorghum Pasta: Composition and Sensory Evaluation with Different Sorghum Hybrids

**DOI:** 10.3390/foods11193124

**Published:** 2022-10-08

**Authors:** Lívia de Lacerda de Oliveira, Lícia Camargo de Orlandin, Lorena Andrade de Aguiar, Valeria Aparecida Vieira Queiroz, Renata Puppin Zandonadi, Raquel Braz Assunção Botelho, Lúcio Flávio de Alencar Figueiredo

**Affiliations:** 1Department of Nutrition, Faculty of Health, University of Brasília (UnB), Campus Darcy Ribeiro, Asa Norte, Brasília 70910-900, DF, Brazil; 2Embrapa Maize and Sorghum, Rodovia MG 424, km 65, Caixa Postal 151, Sete Lagoas 35701-970, MG, Brazil; 3Department of Botany, Institute of Biology, UnB, Campus Darcy Ribeiro, Asa Norte, Brasília 70910-900, DF, Brazil

**Keywords:** sorghum pasta, gluten-free products, antioxidant foods, resistant starch, overall liking, grain diversity

## Abstract

Although whole grain (WG) sorghum is affordable and a healthier alternative to gluten-free pastas (GFPa), sorghum diversity requires evaluation for application in pasta. We aimed to develop GFPa using six sorghum hybrids. White commercial flour (WCF) and sorghums with brown (BRS 305 and 1167048), red (BRS 330 and BRS 332), and white (CMSXS 180) pericarp colors. Total phenolic content (TPC), total condensed tannins (TAN), total antioxidant activity (TAA—FRAP and DPPH), resistant starch (RS), cooking properties, texture, and sensory evaluation were carried out in sorghum pasta. The statistical analyses were ANOVA, Tukey and Friedman test, and multiple factorial analyses. Brown sorghum GFPa showed the best results for bioactive compounds (RS (1.8 and 2.9 g/100 g), TPC (69.9 and 42.8 mg/100 g), TAN (16.9 and 9.4 mg proanthocyanidin/100 g), TAA for FRAP (305 and 195 mM Teq/g), and DPPH (8.7 and 9.0 mg/mL)), but also the highest soluble solids loss (8.0 g/100 g) and lower flavor acceptance for BRS 305. BRS 332 was highlighted for its higher flavor acceptance and intermediary phenolics content. The most accepted pasta was obtained with WCF, and the least accepted with the brown BRS 305. Sweetness (SWE), soluble starch (SS), and DPPH were associated with liking. The main negative variables were WG_flavor, brown color, FRAP, sandy surface (SAN), WG_odor, and TAN. Sorghum hybrids of different pericarp colors are feasible for GFPa production, leading to differences in pasta quality. SAN and GRA, associated with disliking in antioxidant-rich GFPa, could be improved by milling process adjustments. Increasing the SS proportion and SWE with flavors can contribute to the balance between liking and nutritional advantages.

## 1. Introduction

Although gluten is restricted to a few grass species (Triticeae tribe) supplying starch (Table 1), caloric intake for celiac and gluten-free (GF) diet enthusiasts is challenging. Many foods employ starch sources with gluten across various meal preparations (breakfast: bread, cakes, and cookies; lunch and dinner: appetizers, pizza, cakes, and pasta). A shortage of GF products (GFP) creates difficulties for celiacs when eating. Based on serologic test results, celiacs represent 1.4% of the world’s population, or 0.7% based on biopsy data [1]. Even small samples or traces of gluten can contaminate a GF meal.

Moreover, the scarce GFP usually offer poor nutritional, sensory, and technological qualities [2]. Traditionally, the primary sources of GF starch flour are rice (*Oryza sativa* L.), potato (*Solanum tuberosum* L.), corn (*Zea mays* L.), and lesser cassava (*Manihot esculenta* Crantz). However, nutritional quality is usually poor, with the exception of corn. As such, it is essential to identify alternative GF starch sources to improve the nutritional quality of GFP, diversifying the source of starch without inflating the prices of these security crops. 

Sorghum (*Sorghum bicolor* (L.) Moench) is a natural, promising, and affordable alternative to producing healthy GF flour and GFP. As with rice and corn, sorghum grain is GF [3]. Additionally, sorghum is an excellent source of bioactive compounds such as dietary fiber [4], phenolic compounds [5,6], and resistant starch (RS—[7]). The lower domestication time in sorghum [8] may explain such grain attributes, which are typically not found in most of today’s produced cereals.

Sorghum flour has a neutral flavor, making its use in different preparations easier than corn [9]. However, condensed tannins found in some sorghum accessions have been considered anti-nutritional for binding to the protein, reducing its digestibility. This undesirable feature is not expected in a cereal crop as a caloric and protein food and feed source. At the same time, increasing evidence shows that tannins are potent antioxidants, even when in complex with proteins [10], making sorghum an exciting ingredient for functionalizing GFPa through incorporating bioactive compounds [11]. Indeed, dietary fiber and high antioxidant activity make some sorghum grain accessions with tannins healthier for obese individuals, diabetics, and those with high triglycerides [12,13]. Tannins act by decreasing food digestion and, consequently, its glycemic index [14]. In addition, the formation of non-covalent interactions of phenolic compounds with starch results in slowly digestible starch and RS, complementing the inhibition of digestive enzymes during cooking [13]. Furthermore, sorghum proteins called kafirins can act as a barrier to hydrolysis, interacting with starch granules and decreasing amylase accessibility.

Regarding sorghum with tannin, it is worth highlighting the study on feeding Wistar rats on BRS 305 flour with a high-fat, high-fructose diet [12]. This diet improved glucose metabolism, modulated adiposity, and reduced liver steatosis and lipogenesis. These results highlight sorghum with tannin as a potent functional food.

Pasta is one of the most popular foods globally for people of different backgrounds and ages. Dishes with pasta are cheap, easy, quick, and economical to prepare. The different types of pasta, with their numerous sauces, are unanimous. Moreover, dry raw pasta has many advantages in terms of storage and transport. Generally, pasta employs wheat flour, which contains gluten. This starch source, however, shows the most remarkable tendency to increase the food price. In this context, sorghum accessions without tannins may be an excellent starch source for GF flours. Additionally, sorghum accessions with tannins contain exclusive antioxidants and positively affect food digestibility and glycemic index. On the other hand, sorghum also creates a significant drawback to food development since phenolic compounds may interfere with proteins and starch interactions in dough formation. 

In the last seven years, studies about the effect of sorghum in GFPa have not exploited the large diversity of sorghum grains and their effect on pasta characteristics [15,16,17,18]. Only one study evaluated the RS, the phenolic compounds, and the antioxidant activity of sorghum GFPa [18] using two varieties of sorghum (white and brown) decorticated.

The technological appropriateness of sorghum used in pasta formulations depends on the grain’s genetics and composition. Grain diversity and variability affect the cooking properties, such as the optimal cooking time (OCT), the water absorption index (WAI), the volume gain (VG), and the soluble solid loss (SSL) in the cooking water [19]. Moreover, the absence of a gluten network impairs viscoelasticity and texture attributes [20], mainly firmness properties, such as the maximum cutting force (MCF) and the work of shear (WSH). In this context, instrumental analyses should be associated with sensory tests. Besides the acceptance test, rapid descriptive analyses (RDA) have recently been conducted with consumers to assess food’s sensory profile. RDA are a semi-rapid, suitable method to provide descriptive information on sorghum pasta, as they are based on ranking instead of rating and therefore eliminate the evaluator training step [21,22]. However, to assess sorghum pasta quality, these characteristics must be balanced with nutritional characteristics, regarding a higher dietary fiber content, a higher amount of antioxidants and antioxidant activity, a lower total energy value (TEV), and a higher amount of RS, which contributes to the lower glycemic index [23]. 

Unlike most starch crops, sorghum has a C4 photosynthetic metabolism, growing in hot and dry environments, demanding low production inputs related to other starch crop sources. With population expansion, limitations on water availability, the temperature rise, and the need for cultivation on marginal lands, sorghum is an additional key crop for human caloric intake and GF diets. Furthermore, sorghum has high diversity, meaning that can be used to develop many healthy products with a variable number of bioactive compounds and nutrients depending on the applied material. Moreover, it is one of the cheapest GF sources. The aim of this study was to evaluate fresh GFPa’s technological, sensory, and nutritional characteristics with six sorghum hybrids with and without tannin, comprising three pericarp colors: white, red, and brown.

## 2. Materials and Methods

### 2.1. Sorghum GRAIN HYBRIDS

Embrapa Maize and Sorghum provided sorghum flours of five hybrids (BRS 305, BRS 330, BRS 332, 1167048, and CMSXS 180). Embrapa developed these hybrids, with good performance in yield. The sixth sorghum flour was a commercial white sorghum flour (Bob’s Red Mill, Milwaukie, OR, USA). The sorghum grain pericarp colors were white (commercial and CMSXS 180), red (BRS 330 and BRS 332), and brown (BRS 305 and 1167048). The brown sorghum had a pigmented testa layer. The same flours, except commercial flour, were analyzed regarding their composition and antioxidant content [5].

### 2.2. Pasta Development 

We initially adapted the GF psyllium pasta [24] without sorghum flour. Our dough contained 24.4% sorghum flour, 12.2% potato starch (Yoki, General Mills, Brazil), 12.2% rice flour (Urbano, Jaraguá do Sul, Santa Catarina, Brazil), 34% egg white, 11.4% water, and 5.8% psyllium. This latter was obtained from a manipulation pharmacy in Brasilia, Federal District, Brazil, with a granulometry passage of 97.85% on 60 meshes (250 microns), with an apparent density of 0.57 g/mL, according to the supplier (Medicare DF). The dough was kneaded with a hand roller and left to rest for 10 min. After this period, samples were folded and passed through a manual pasta machine (Cuisinart Brasil, São Paulo, Brazil), and, later, they were cut into noodles 15 cm in length and 0.5 cm in width. The pastas were left to dry for 12 h on a drying line and then stored under refrigeration at 4 °C for 48 h. For chemical analysis, the pasta was cooked for 15 min in boiling water (100 g pasta in 1 L of water), frozen, and lyophilized. Therefore, we used freshly cooked pasta for chemical, sensory, texture, and descriptive analysis.

### 2.3. Nutritional Composition

The nutritional composition was estimated by the combination of the centesimal composition of the same hybrids of sorghum flour [5] on GF bread production, the Brazilian Table of Food Composition [25], and the Brazilian Table of Food Composition of the University of São Paulo (USP, 2008) for ingredients such as rice flour, potato starch, and egg white. The Atwater coefficients were calculated the total energetic value (TEV) using 4.0 for proteins, 4.0 for carbohydrates, and 9.0 for lipids [26]. Calculations were adjusted to cooked pasta using the water absorption index determined in Section 2.5.

### 2.4. Chemical Analysis

#### 2.4.1. Soluble and Resistant Starch (SS and RS)

The determination of RS used the Megazyme K-SRTAR 09/14 kit according to AOAC Method 2002.02 [27]. The combination of pancreatic α-amylase (10 mg/mL) and amyloglucosidase (3 U/mL) was used to perform the enzymatic hydrolysis of soluble starch (SS) at 37 °C for 16 h [7]. SS was separated in the supernatant and pellet by centrifugation. The latter contained the RS, washed with ethanol and solubilized with 2 mol/L KOH. The concentration of RS was measured at 510 nm and expressed as g/100 g sorghum flour.

For the quantification of SS, the volumetric flasks containing the supernatants obtained from enzymatic hydrolysis were subjected to subsequent washes with ethyl alcohol and mixed thoroughly, and volumes were enhanced to 100 mL with 0.1 mol L^−1^ sodium acetate buffer (pH 4.5). The same procedure as used for hydrolysis with amyloglucosidase for absorbance readings at 510 nm was conducted to assess the total SS.

#### 2.4.2. Amylose Content (AM)

The amylose/amylopectin content was determined using the Megazyme (Ireland) kit K-AMYL 09/14 by the dimethyl sulfoxide (DMSO) method. The GFPa samples had their amylose content determined enzymatically and analyzed using glucose/peroxidase reagent (GOPOD).

#### 2.4.3. Extract Preparation for Antioxidant Analyses

In a 15 mL falcon tube, 0.5 g of sample was weighed, and 10 mL of HCL:methanol (70%:1%) was added; then, the sample was placed on a shaker table for 2 h and kept at rest overnight. The mixture was centrifuged at 4000 rpm for 10 min, and the supernatant was collected in a 50 mL falcon tube. The extraction residue was added to 10 mL of a 70% methanol solution acidified with 1% HCl. The sample was homogenized on the shaker table for 30 min. After further centrifugation, the supernatant was added to the 50 mL Falcon tube, extracted twice with 10 mL of 70% acetone. The supernatant was transferred to a 50 mL volumetric flask, and distilled water was used to complete the volume. The extract was filtered with glass wool and kept in an amber glass at −80 °C.

#### 2.4.4. Total Phenolic Content (TPC)

The Folin—Ciocalteu method [28] was carried out, and the results were expressed as mg gallic acid/100 g.

#### 2.4.5. Total Condensed Tannins (TAN)

TAN were analyzed according to 4-dimethylaminocinnamaldehyde (DMAC) [29], using 70 µL from each extract and adding 210 µL of DMAC reagent in a polystyrene microplate with 96 cavities. The microplate was installed in the spectrophotometer, and after 25 min, we read the absorbance at 630 nm. Results were expressed as mg proanthocyanidin B2.mL^−1^.

#### 2.4.6. Total Antioxidant Activity (TAA)

TAA by 2,2-diphenyl-1-picryl-hydrazyl-hydrate (DPPH) was determined using the DPPH method [30], modified slightly [31]. TAA was assessed by the inhibitory concentration (IC50), with the results expressed in mg/mL.

TAA by ferric reducing ability power (FRAP) [32] was carried out by transferring a 90 μL aliquot of each extract to tubes in three consecutive dilutions. Previous tests used sample concentrations of 100, 75, 50, 25, and 12.5%. Results were expressed as µmol TE.g^−1^.

### 2.5. Pasta Properties

The optimum cooking time was determined by the AACC method 66-55 [33]. After setting the cooking time in minutes, we evaluated the pasta at the optimal cooking time for the increase in water absorption index (g/100 g), VG (mL), and SSL in the cooking water (g/100 g).

The water absorption index [33] was determined by the weight gain and the difference in weight of the raw sample (10 g of fresh dough) and after cooking for the optimal time.

The volume gain (mL) was measured before and after cooking. Pastas were immersed in 140 mL of hexane, and subtraction between the volume displaced by the hexane when raw and cooked was used to calculate the value of the volume gain.

Cooking loss (g/100 g) was determined through the SSL in the cooking water according to the AACC method 66-55 [33].

### 2.6. Texture Analysis

The TA-XT plus texture analyzer (Stable Micro Systems, Surrey, UK) was used, with a light knife blade (A/LKB) probe and a load cell of 5 kg. The protocol was carried out with a speed of 0.5 mm/s for pre-tests, 0.17 mm/s for the test, and 10 mm/s for the post-test. This method measured the force required to cut five strands of pasta positioned adjacent to one another. The test produced a force versus distance recording, where the area under the curve was the WSH required for shearing the cooked pasta, expressed in g.cm. The MCF was the peak force required to cut the sample (g), referring to its resistance to breakage. Thus, firmness was assessed by force required to bite (MCF) and pasta resilience (WSH). 

### 2.7. Sensory Analysis

#### 2.7.1. Acceptance Test

The analysis took place in the Sensory Analysis Laboratory of the University of Brasilia. Acceptance was assessed with the freshly cooked pasta using a 9-point structured hedonic scale and 121 consumers between 18 and 75 years old. Females comprised 78% of students and staff of the University of Brasilia. Each pasta sample of 15 g with a simple tomato sauce was coded with a three-digit code and served in balanced and randomized order, between 65 and 70 °C. Water was provided to rinse the palate before tasting each sample. A survey was handed out to all the participants with questions about food allergies, pasta consumption, and GFPa consumption before the tests.

#### 2.7.2. Descriptive Analysis

The RDA [21,22] was carried out for sensory characterization. The procedure is similar to a conventional descriptive profile, except that assessors are not trained in intensity scales after descriptor elicitation. There is a single contact with reference pastas for basic calibration, and pastas are ranked instead of rated [21]. We recruited 31 assessors among nutritionists working in the food production sector in Brasilia. They compared the six sorghum GFPa by ranking them according to each attribute, which the same group previously elicited with the repertory grid procedure [34]. Descriptors were elicited from the six sorghum GFPa, by comparison of differences and similarities among them using the triadic elicitation procedure. 

### 2.8. Experimental Design and Statistical Analysis

Experiments were carried out in a completely randomized design with six treatments (sorghum flours) and three replicates. Each experimental unit was a package with 500 g of pasta. ANOVA analyzed data, and the Tukey test (*p* < 0.05) was applied if significant. A complete randomized block design analyzed the sensory data, and acceptance data were subjected to ANOVA and Tukey test before and after cluster analysis. Comparisons were carried out between and within consumers’ clusters. RDA data were analyzed by the Friedman test (*p* < 0.05). Principal component Analysis (PCA) was applied on sorghum GFPa and their respective flours’ [5] instrumental characteristics. Multiple factor analysis (MFA) was run between instrumental and sensory data from RDA. 

The University of Brasília Ethics Commission approved the research (CEP/FS CAAE 51021415.0.0000.0030). 

## 3. Results and Discussion

We developed sorghum GFPa (Figure 1), which has a better functional composition than other commercial GF starch sources and could be more affordable than other GFPa. This study joined the six other scarce essays offering sorghum-based GFPa with nutritional/chemical, sensory, and technological properties, such as pre-heated, extruded, and dried noodles [35] and dry pasta [15,16,17,18,36,37]. Besides focusing on fresh pasta, our work differs from the others by the higher proportion of sorghum flour as a starch source used. Furthermore, our study is the first that uses more than two sorghums to prepare sorghum pasta, allowing multivariate analysis to understand possible correlations between composition and quality traits. 

Regarding the nutritional composition, the TEV of GFPa ranged from 86.26 (red BRS 330) to 98.03 Kcal/100 g (bronze 1167048), with a small variation of 13% among the hybrids (Table 2). The pasta produced with brown sorghum (BRS 330) presented the highest WAI (3.66—Table 3), leading to macronutrient dilution and decreasing TEV. In the best GFPa obtained from sorghum, rice, and potato flours (40:20:40—[15]), the TEV of the cooked pasta was 129.65 Kcal/100 g, with 1.51 g/100 g of lipids. Since we did not use oil or egg yolk, our lipid content was lower (from 0.23 to 0.43 g/100 g). PR ranged from 1.77 to 2.04 g/100 g in our cooked pasta. White (Pannar-8706 W) and brown (Pioneer-81G67) sorghum cooked pasta [17,18] presented from 3.6- to 5.3-fold more protein than our pasta, after the conversion of our data to a dry basis for suitable comparison. This can be explained by the use of protein-rich ingredients such as egg powder and egg albumen in their formulation [18]. On the other hand, fiber content varied from 2.54 (BRS 330) to 3.42 g/100 g (BRS 332), which is within the range presented for the white and brown sorghum cooked pasta [18] in comparable bases. The use of 5.8% of psyllium flour contributed to fiber enrichment in pasta formulations.

CHO in fresh cooked pasta ranged from 19.12 (BRS 330) to 21.32 g/100 g (CMSXS 180). The main carbohydrate in our GFPa was SS, ranging from 9.5 g/100 g on brown 1167048 to 12.7 g/100 g on red BRS 330 GFPa. RS varied 24 times between the red BRS 332 (0.2 g/100 g) and the brown 1167048 (4.8 g/100 g) and the red BRS 330 (4.6 g/100 g). These data indicated variability within the same pericarp colors for both red hybrids (BRS 330 and BRS 332). Moreover, RS was also different between the browns (4.8 for 1167048 versus 3.4 g for BRS 305) and the whites (1.8 for CMSXS 180 and 0.6 g for white commercial sorghum GFPa).

The percentage of amylose was more homogeneous with pericarp colors, except for brown sorghum pasta. Indeed, BRS 305 showed the highest percentage of AM, with 36.1%, followed by the proportion found for white sorghum pasta (17.6 to 18.8%), and finally the red ones, besides the other brown sorghum GFPa 1167048 (14.3 to 14.7%). AM is reported to increase firmness and resilience in pasta. The optimum level of amylose for better pasta cooking quality for wheat flour was estimated to be 32 to 44% [38]. We previously reported that AM in the flour of these five hybrids ranged from 11.50 (CMSXS 180) to 22.75% (1167048) [5]. The most remarkable change in the proportion of AM from flour to cooked pasta was observed for BRS 305, reaching 36%.

Different proportions of leached AM and amylopectin content (AP) were found in the poured cooking water of rice grains [39]. At 85 °C, rice cooking water presented 31.55% of AM and 63.27% of AP. Besides temperature, other factors in the inner composition can influence the proportion of leached AM and AP in the cooking water. A study on the interaction of sorghum proanthocyanidins (mainly constituted by tannins) with starch molecules after cooking showed that tannins interact differently with the AM and AP. The interaction appeared to be specific to amylose and linear fragments of amylopectin, probably involving hydrophobic interactions. The high proportion of hydrophobic sites provided by the physical conformation of the polymeric proanthocyanidins interact with the hydrophobic core of the linear helical structure of amylose, which is more accessible in solution compared to amylopectin. Indeed, amylopectin side chains provide limited hydrophobic sites, besides the steric hindrance that would interfere with its ability to efficiently interact with the polymeric tannins [40]. In our recent study with GF bread [5], the BRS 305 sorghum flour presented the highest amount of TAN (609.9 mg PE/100 g FW), which remained expressive among the sorghum GFPa. It is suggested that the interaction of TAN and AM in this sorghum contributed to the reduced leaching of this starch fraction, making AM rise in the cooked pasta concerning the other hybrids. AM in cooked pasta, driven by the interactions with tannins, seems to influence pasta texture, as the peak of viscosity of starch measured in the Rapid Visco Analyzer (RVA) increased with the content of sorghum tannins [40]. A PCA on the chemical composition of sorghum flours and physical, chemical, and technological characteristics of the cooked pasta (Appendix A) [5] showed a significant correlation between AM and TAN using flours (*r* = 0.773) and cooked pasta (*r* = 0.815). These correlations and the positions of variables in the PCA map corroborate the AM retention in cooked sorghum GFPa with higher TAN.

The addition of sorghum flour to pasta increases its RS content significantly [18,41]. Palavecino et al. [41] obtained 1.36 g of RS when adding 40% red sorghum and 1.16 g when adding 40% white sorghum flour, while the control pasta presented 0.39 g/100 g. These values are slightly lower than ours (from 0.2 to 4.8%—Table 2) for some hybrids, but the amount of sorghum flour added was 25% lower than in our study (50% of total flour). The red BRS 332 and the white commercial sorghums showed a low RS content (0.2 and 0.6%, respectively—Table 2). Still, these values could be explained by the growing conditions of sorghum or the phenotypic differences [42].

The GFPa with TAN using the brown sorghum BRS 305 showed the highest TPC (69.9 mg/100 g), differing statistically from the other five sorghum flour hybrids used (25.2 to 42.8 mg/100 g—Table 2). All these pastas were prepared with 24.4% of sorghum flour. On the other hand, white (WSP) and brown (BSP) sorghum pasta using 62.7 and 79.8% of sorghum flours, respectively [17], showed 241 and 288 mg/100 g of TPC [18]. The approximately three-times higher amounts of sorghum flour may explain these high TPCs.

The TAN of pasta from brown pericarp grains (1167148 and BRS 305) had higher averages (9.4 and 16.9 mg/100 g, respectively—Table 2), differing from the other sorghum GFPa (from 3.5 to 5.7 mg/100 g). One serving of 100 g of BRS 305 sorghum pasta presents tannin content comparable to that of Merlot red wine (14.69 mg/100 g) [43]. None of the six studies [15,16,17,18] of sorghum GFPa evaluated TAN in their samples. However, a large sampling of sorghum (*n* = 287 lines and hybrids) with different pericarp colors recorded a TAN variation from 0 to 101 mg/100 g, with an average of 7.1 (validation set, *n* = 63) and 14.8 (calibration set, *n* = 224) [44]. Considering our GFPa with 24.4% of sorghum flour and the processing losses, it is possible to verify the good retention of TAN in the cooked pasta.

TAA by DPPH expressed as IC 50 represents the inhibitory concentration or the amount of antioxidant extract necessary to inhibit a certain concentration of a radical. Higher values indicate lower antioxidant activity. The brown sorghum pastas presented higher TAA by DPPH, followed by the red ones (BRS 330 and BRS 332), the white CMSXS 180, and the white commercial sorghum pasta. The values of DPPH IC 50 ranged from 8.7 mg/mL to 27.5 mg/mL dry basis (Table 2). The DPPH IC 50 of decorticated sorghum grains ranged from 0.24 to 0.50 mg/mL in comparison with 0.05 mg/mL for Trolox [45]. Considering flour and pasta processing and dilution into the total formulation, the antioxidant capacity of sorghum pasta in our study is lower (from 17.4- to 114.5-fold) than those of sorghum grains. The DPPH IC 50 values of sorghum flours of the same hybrids used in this study were expressed on a fresh basis, ranging from 6 to 22 mg/mL [5].

For the FRAP method, the best TAA was found for brown sorghum pastas but was significantly higher for BRS 305 than 1167048. No difference was found between red and white sorghum counterparts. Values ranged from 34 to 305 mM TE/g dry basis (Table 2). The values of TAA in the sorghum flours of these hybrids ranged from 63 µM TE/g for CMSXS 180 to 802 µM TE/g fresh basis for BRS 305 [5], indicating a good carry-over of antioxidant capacity again after the formulation and processing of sorghum pasta.

Pasta with red sorghum (BRS 330) was statistically higher in WAI and SSL than white (CMSXS 180 and commercial) and the other red one (BRS 332). However, there was no statistical difference among the six pasta hybrids for VG (Table 3). The classification of pasta quality regarding the SSL is as follows: (i) excellent quality when it loses up to 6% of solids, (ii) good quality when it loses up to 8%, and (iii) low-quality loss when it loses equal to or more than 10% [46]. Therefore, the SSL (from 6.3 to 7.9%—Table 3) of our pasta was good or very good, and they were worse than GFPa (from 0.85 to 1.10%) with grey tannin-free sorghum, rice, and potato flours [15]. On the other hand, SSL was close (from 7.6 to 8.4%) to GFPa with white tannin-free sorghum and rice flour [16] and better than GFPa with white and brown sorghum (from 3.68 to 11.94%—[17]).

The WAI in traditional gluten pasta varies from 1.3 to 2.5 [24], lower than in this study (from 3.18 to 3.66—Table 3). On the other hand, the water absorption for the sorghum pasta obtained in another study [17] ranged from 1.3 to 1.9. This formulation contained a higher amount of egg powder and egg albumen, conducted on a more compact network that prevented the water uptake of the other ingredients [17]. 

The long cooking time caused the tremendous loss of soluble solids in water [24]. We observed that the wheat standard pasta had less SSL because of its gluten network. The sorghum flour and psyllium fibers partly replaced the gluten network, but it was not enough for the BRS 330 pasta, thus leading to the higher leaching of compounds (7.97%—Table 3) in the cooking water.

Texture evaluation was based on the maximum force required as a bite or cutting and the work required to achieve it (the area under the curve), which indicates its resilience [47]. The values of MCF of sorghum pastas were between 648 and 907 g (Figure 2). These values are higher than those found for wheat/rice noodles (235 to 357 g—[48]), indicating that the force required in the first bite is higher for sorghum pastas. Higher values of 4606 and 5384 g, for WSP and BSP, respectively, were recorded for firmness in TPA—peak force in the first cycle of compression [17]. The use of different probes (cylinder) and methods (TPA) contributes to differences. Moreover, they used several ingredients related to higher dough viscosity (xanthan gum, egg albumen, egg powder, pregelatinized starch), whereas we used psyllium. Extrusion and drying were processing operations that also contributed to the higher peak forces in that study. As another reference, a commercial wheat tagliarini pasta with the same equipment and protocol presented an MCF of 533 g (personal data), showing that the use of sorghum flour in pasta increases the MCF in comparison with wheat flour.

The red BRS 332 sorghum pasta showed the highest MCF (907 g), followed by commercial sorghum pasta (823 g). Red BRS 330, white CMSXS 180, and brown BRS 305 products were similar regarding MCF (from 673 to 735 g). In contrast, the brown 1167048 sorghum pasta presented the lowest value of MCF (648 g) but was statistically similar to its brown sorghum pasta counterpart. WSH ranged from 78 to 95 g.cm in our study (Figure 2), which was higher than the WSH of 15.6 to 28.2 g.cm of wheat rice noodles [48] and higher than the WSH of 41 g.cm of wheat tagliarini pasta (personal data), indicating the higher resilience of sorghum pastas. The highest values of WSH were for the red BRS 330 and the brown 1167048 sorghum pastas. Any differences appeared among the other pastas. Divergences in the firmness of cooked pasta are primarily due to gluten [47]. In the case of GFP, other components such as proteins, amylose, fiber, resistant starch, or tannins may interfere with these products’ structures and textural properties. Further, the multivariate analysis could explain them better. It is worth noting that MCF and WSH did not show the same trend, revealing different mechanical aspects of the pasta.

The position of the peaks indicates that the blade is cutting through tougher or stronger layers than weaker ones. The first peak indicates the entry of the blade into the sample. If the material caves in only slightly, the penetration force is low, and the drop after blade entry is also low, indicating a lower maximum cutting force but a higher work of shear. On the contrary, there is a large cave in some food materials before blade entry, giving a high peak force and usually a significant drop in force in the inner layers (https://textureanalysisprofessionals.blogspot.com/2018/01/physical-property-measurement-fracture.html accessed on 23 September 2022).

The PCA (Appendix A) of sorghum flour’s chemical and physical characteristics in GFPa elucidated interesting associations with MCF and WFH. The highest and most significant correlations of WSH were, in descending order, RS (*r* = 0.804), AM (*r* = 0.735), and MOI (*r* = −0.726), whereas MCF correlated with RS (*r* = −0.787), TAN (*r* = −0.622), and insoluble fiber (*r* = −622). The second dimension of the PCA map represents the proximity of WSH with RS and AM in the flour and a greater distance to soluble fiber and moisture, suggesting a central role of these variables in the textural properties of sorghum GFPa. In general, the sensory acceptance of sorghum pasta (Table 4) was low, as expected for a food that is not part of traditional eating habits, such as sorghum pasta, which suffered a notable change from the standards when replacing the traditional wheat flour with GF starch sources. The effect of partial replacement (20 to 40%) of durum wheat flour with white and red sorghum flours in pasta found acceptance close to 5.0 or 6.0 [41] on the 9-point hedonic scale, similar to the means found in this study for sorghum GFPa. Another study [17] showed acceptance ranging from 3.8 (brown sorghum pasta) to 4.5 (white sorghum pasta). GFPa were developed with 100% sorghum flour, 100% corn flour, and 50–50% sorghum–corn flours, and the acceptance of the pasta was evaluated by celiac and non-celiac subjects [36]. The acceptance means of sorghum pasta were lower than 5.0 when considering non-celiac people, which were slightly lower than the means achieved in our study. On the other hand, sorghum pasta acceptance was close to 7.0 for the celiac group in that study.

Comparing sorghum accessions helps to evaluate their potential applicability to GFPa production. Commercial white flour pasta was more accepted than brown BRS 305 pasta, whereas the other treatments showed intermediate acceptance. BRS 305 pasta was also the least accepted for flavor. Pasta from bronze pericarp (BRS 332) had more acceptance than the other bronze (BRS 330) pastas, and they did not differ from the white (commercial and CMSXS 180) pasta or brown (1167048) pasta. The acceptance of color and odor was similar for all GFPa. 

Cluster analysis for the overall impression data of the 121 assessors checked the possible groups with different behaviors concerning pasta acceptance. Cluster 1 (41 consumers) differed from cluster 2 (80 consumers) by showing higher acceptance for all pastas considering all the evaluated attributes. Cluster 1 was also characterized by liking equally all sorghum pastas. On the other hand, cluster 2 was more discriminative since they rejected more pastas made with brown pericarp grains (BRS 305) than white and red pericarp grain (CMSXS 180 and BRS 332) pasta, regarding flavor. Similarly, for overall impression, cluster 2 preferred the white sorghum pastas in comparison with the same brown BRS 305 GFP.

Sensory descriptive analysis (Table 5) elicited eight attributes comprising appearance (brown color—BROWN, sandy surface—SAN), odor (whole-grain—WGO and pasta odor—PO), flavor (sweetness—SWE and whole-grain flavor—WGF), and texture (hardness—HAR and granulosity—GRA). For example, Andean corn pasta was described in a descriptive method as rough, with dark spots, and a strange odor and flavor [49]. These pastas were also sandy, less yellow, had an odd flavor, corn-like flavor, cereal-like smell, and the presence of specks, and less egg odor and flavor. 

Hardness was the only attribute that showed no difference among the GFPa. The BROWN attribute was rated very coherently from white to brown sorghum flours. The pairs of white, bronze, and brown GFPa did not differ. SAN and GRA are probably related attributes and associated with the particle sizes of the flours. Indeed, commercial white flour showed lower intensity for these attributes, and only the red BRS 330 pasta was similar to this sample for GRA. The brown BRS 305 pasta was found to be more granular than the bronze BRS 332 pasta, and the other pastas were intermediate. SAN was also more intense for BRS 305 pasta than the bronze pastas. The difference in the particle sizes of the sorghum flours for the same milling process is generally related to the endosperm texture of each accession, which varies between farinaceous and vitreous [50,51]. In our previous study [5], the endosperm texture of BRS 305 was more farinaceous than BRS 332. Large particles tend to under-hydrate, resulting in a stiff dough and a higher tendency to crack [11]. Moreover, particle precipitation formed by the interaction between tannins and proteins can also lead to astringency and a sandy sensation in the mouth [52].

According to MFA (Figure 3), in which qualitative and instrumental results are related, in the first dimension (53% of explanation), the clusters’ acceptances (OLC1 and OLC2) are associated with the descriptive SWE obtained by RDA and MCF, and the lower antioxidant activity represented by higher DPPH-IC5O. The factor loadings in the first dimension indicated that SWE seems more associated with the lower content of TPC and TAN than the higher content of CHO and SS. The positive correlation with DPPH IC 50 indicated that pasta with lower antioxidant activity was more acceptable. The maximum cutting force (MCF) trait was also related to acceptance. MCF is the load observed before breaking, associated with higher bite resistance. On the other hand, the WSH is more associated with the resilience of inner layers to collapse. The heterogeneous granulometry of the mass possibly caused the distancing of the results for MCF and WSH. It is worth noting that WSH was positively associated with descriptive hardness (HAR) and negatively associated with sorghum classified as red or white pericarp and without the presence of tannins (commercial, CMSXS 180, BRS 332, and BRS 330) but positively associated with TAN, TPC, RS, and AM. These chemical characteristics were also associated with higher OCT. The difference in the red sorghum pasta is highlighted in the second dimension (22% of explanation) of the MFA map. BRS 332 was more correlated to FIB, PR, CHO, and ASH, in one pole, besides MCF, whereas, in the other pole, BRS 330 indicated a lower content of these macronutrients and a higher WAI, SSL, WSH, and SS.

The descriptive variables of GRA, HAR, SAN, BROWN, WGF, WGO, and PO, besides the instrumental variables of RS, TPC, FRAP ATT, and TAN, and the texturometric WSH and MCF, are in the opposite quadrant to acceptance (Figure 3). This observation indicated a rejection by the evaluators associated with these variables. It is necessary to highlight the perception of the sweeter tastes by evaluators. Therefore, components such as phenolics, including tannins, compromise the palatability of the products and lead to negative consumer responses since they are related to astringency and bitterness in products [53]. These variables were positively associated with sorghum accessions with brown pericarp and the presence of tannins (BRS 305 and 1167048).

Correlations between texture (WSH, SSL, WAI, and VI) and chemical variables (TPC, TAN, RS, and AM) could explain how these variables influence pasta structure. The role of AM in firmness and resilience [38] is emphasized by the map’s close positioning of AM, WSH, and HAR. Commercial sorghum pasta was emphasized as the preferred product, followed in the first dimension by red BRS 332 and white CMSXS 180 sorghum pastas. The assessors in the sensory analysis did not have a habit of consuming whole grains and GFPa (stated by previous questionnaires) but were conventional pasta consumers. Thus, characteristics such as browning, taste, and odor in whole products were also negatively correlated to the sensory acceptability of brown sorghum pasta. Therefore, further studies must evaluate the effect of consumers’ habits in consuming whole grains on the acceptance of products made with tannin-rich brown sorghums. In addition, the attributes of GRA and SAN are related to the granulometry of sorghum flour used in the preparation of the pasta (Table 5).

SS was associated with WAI, but not VI, and less to SSL (Table 3). In addition, the sorghum GFPa did not have high PR. Therefore, it probably did not form a strong network in the dough, so the starch polymers were less trapped in the food matrix, increasing the SSL [54], representing the high solubility of the starch used in the formulation. Increasing PR is an additional strategy to optimize these formulations in this context. Additionally, transglutaminase has been reported as another alternative to building protein networks in GF doughs [55]. Regarding the process, a feasible strategy could be process optimization in terms of mixing (time, temperature, and speed rate) parameters to improve textural, sensory, and technological adjustments [11]. 

FIB was negatively associated with WAI and SSL. The type of fiber is crucial in the interactions with water. Sorghum grains present a higher proportion of insoluble fiber [5], which was partially balanced with the content of soluble fiber provided by psyllium flour. Thus, WAI and the SSL were more explained by the content of SS in our formulation.

Grain decortication (dehulling) looks to eliminate the pericarp, keeping only the predominant starch in the endosperm, which is involved in the final product quality. This grain bran is rich in fiber and bioactive compounds for human health [56]. With sorghum and millets, grain decortication is common in Africa and Asia and comprises at least 5 to 20% of the whole grain [57,58,59]. Undernutrition is a significant drawback in these continents. As such, grain dehulling and other grain processing factors eliminate phenolic compounds such as tannins, increasing starch and protein digestibility. This rich bran is usually linked to animal feed. On the other hand, red and brown sorghum grains seem to maintain some of these bioactive compounds after being decorticated [18,45].

Unlike many GFP developed, we prioritize the use of whole sorghum grain for breads [5] and pastas. The essential and rare functional compounds present in the pericarp are usually absent in GFP. In 2016, the world adult (>18 years) population includes 39 and 13% of overweight and obese people, respectively (https://www.who.int/news-room/fact-sheets/detail/obesity-and-overweight accessed on 3 October 2022). Therefore, sorghum GFPa can be helpful for celiacs and other essential diets that address overweight, obesity, diabetes, and triglycerides, among other chronic diseases. At the same time, our formulations aimed to reduce the negative impacts of pericarp on the quality of the final product. Our results are also relevant by pointing to the necessary adjustments that could lead to sensory improvements in antioxidant-rich sorghum pasta, thus optimizing the nutritional and sensory aspects of GFPa formulations. 

## 4. Conclusions

This study aimed to give more options and quality improvements for pasta-based GF diets. Thus, six types of pasta were developed and analyzed with 50% of the starch source from six sorghum hybrids—two of them with TAN. Thanks to sorghum, these GFPa were good sources of functional compounds such as resistant starch and phenolic compounds, which lead to higher antioxidant activity. The hybrid BRS 305 (brown pericarp with tannins) achieved satisfactory results for these traits, the absorption rate, and the increase in volume. 

Pasta using brown sorghum pericarp flours (BRS 305 and 1167048) showed several nutritional advantages. However, phenolics’ impairment of sensory quality, especially tannins, remains a significant drawback and requires further studies to improve the formulation and processing, mainly for BRS 305 pasta. Our study pointed to crucial strategies, such as reducing GRA and flavor strategies, to improve SWE and increase SS in the other 50% of starch sources, as this variable was associated with SWE and liking. Moreover, increasing the protein content and additives could bring texture and structural ameliorations. On the other hand, red sorghum flours showed excellent potential in pasta production as they presented intermediary amounts of antioxidants and acceptance similar to that of white commercial sorghum flour. The future evaluation of acceptance by specific consumers of whole-grain products or celiac groups could include the acceptance of GFPa richer in antioxidant compounds from brown sorghum flours by a more specific target audience.

## Figures and Tables

**Figure 1 foods-11-03124-f001:**
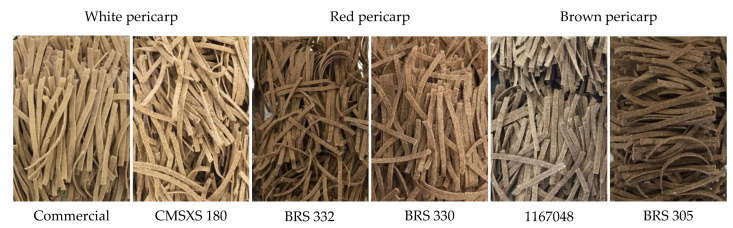
Sorghum pastas used six hybrids with three pericarp colors.

**Figure 2 foods-11-03124-f002:**
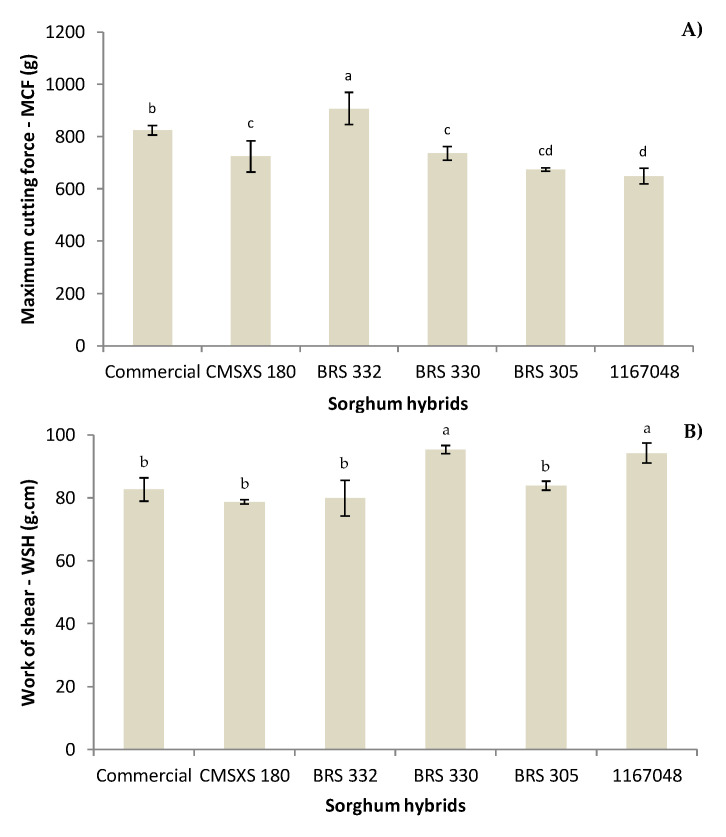
Texture analysis of GFPa produced with sorghum flour as 50% of the starch source: (**A**) maximum cutting force (MCF); (**B**) work of shear (WSH). Bars with the same letters do not differ significantly (*p* > 0.05).

**Figure 3 foods-11-03124-f003:**
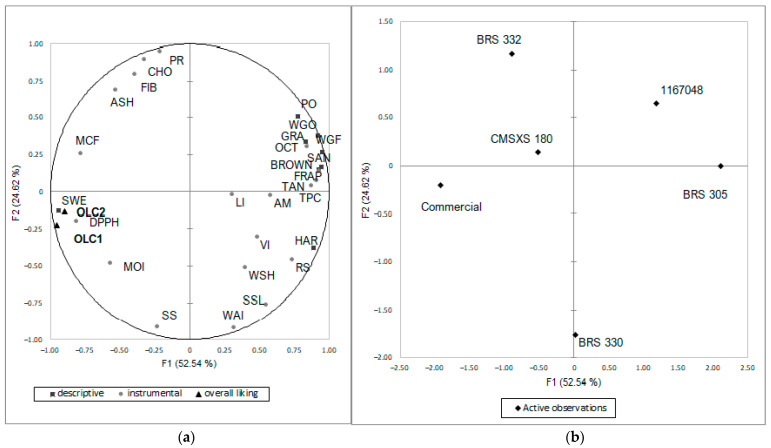
(**a**) MFA map. (**b**) Accession observations. Variables are: (1) Acceptance: OLC1—overall liking cluster 1; OLC2—overall liking cluster 2; (2) Descriptive: SWE—sweetness; HAR—hardness; GRA—granulosity; SAN—sandy surface; BROWN—brown color; WGO—whole-grain odor; WGF—whole-grain flavor; PO—pasta odor; (3) Instrumental: CHO—carbohydrates; PR—protein content; LI—lipid content; FIB—fiber; ASH—ash; TPC—total phenolic content; TAN—total condensed tannins; DPPH—antioxidant activity EC50; FRAP—antioxidant activity; RS—resistant starch; AM—amylose content; SS—soluble starch; MOI—moisture; WSH—work of shear; MCF—maximum cutting force; OCT—optimum cooking time; WAI—water absorption index; VI—volume increase; SSL—soluble solids loss.

**Table 1 foods-11-03124-t001:** Primary caloric sources for humans from starch with and without gluten.

Starch Sources	Name	Scientific Name	Family/Subfamily/Tribe	Starch Bearing Tissue
Gluten-free	Cassava Maize (corn) Finger millet Pearl millet Potato Quinoa Rice Sorghum Teff	*Manihot esculenta* Crantz *Zea mays* L. *Eleusine coracana* Gaertn. *Cenchrus americanus* (L.) Morrone *Solanum tuberosum* L. *Chenopodium quinoa* Wild. *Oryza sativa* L. *Sorghum bicolor* (L.) Moench *Eragrostis tef* (Zucc.) Trotter	Euphorbiaceae/Euphorbioideae/Manihoteae Poaceae/Panicoideae/Maydeae Poaceae/Chloridoideae/Cynodonteae Poaceae/Panicoideae/Paniceae Solanaceae/Solanoideae/Solaneae Amaranthaceae/Chenopodioideae/Chenopodieae Poaceae/Ehrartoideae/Oryzeae Poaceae/Panicoideae/Andropogonae Poaceae/Chloridoideae/Eragrostideae	Root Grain Grain Grain Stem Grain Grain Grain Grain
Gluten	Barley Oat Rye Triticale ^1^ Wheat Wheat	*Hordeum vulgare* L. *Avena sativa* L. *Secale cereale* L. *× Triticosecale* Wittm *Triticum aestivum* L. (bread) *T*. *turgidum* subsp. *durum* Desf. (pasta)	Poaceae/Pooideae//Triticeae Poaceae/Pooideae/Aveneae Poaceae/Pooideae/Triticeae Poaceae/Pooideae/Triticeae Poaceae/Pooideae/Triticeae Poaceae/Pooideae/Triticeae	Grain Grain Grain Grain Grain Grain

^1^: Triticale is a hybrid of wheat (female parent) and rye (male parent).

**Table 2 foods-11-03124-t002:** Chemical composition of cooked sorghum GFPa.

Genotypes	TEV	CHO	PR	LI	FIB	ASH	MOI	SS	RS *	AM	TPC*	TAN *	DPPH *	FRAP *
Commercial (white)	95.3	21.21	1.88	0.33	3.00	0.39	14.6 ^b^ ± 0.1	11.2 ^b^ ± 0.1	0.30 ^e^ ± 0.02	18.8 ^b^ ± 1.2	25.2 ^d^ ± 0.4	3.50 ^c^ ± 0.2	27.5 ^a^ ± 0.22	33.6 ^c^ ± 2.8
CMSXS 180 (white)	95.4	21.32	1.92	0.27	2.73	0.39	14.5 ^bc^ ± 0.3	10.8 ^b^ ± 0.0	0.94 ^d^ ± 0.09	17.6 ^b^ ± 0.4	33.2 ^c^ ± 0.9	4.75 ^c^ ± 0.3	25.7 ^b^ ± 1.02	45.2 ^c^ ± 3.5
BRS 332 (red)	97.0	21.70	2.04	0.23	3.42	0.44	14.9 ^ab^ ± 0.3	10.3 ^c^ ± 0.2	0.12 ^f^ ± 0.02	14.3 ^c^ ± 0.6	42.0 ^b^ ± 1.5	5.00 ^c^ ± 0.1	22.3 ^c^ ± 0.38	60.3 ^c^ ± 3.4
BRS 330 (red)	86.3	19.12	1.77	0.30	2.54	0.36	15.4 ^a^ ± 0.1	12.7 ^a^ ± 0.0	2.05 ^b^ ± 0.03	14.7 ^c^ ± 0.5	41.9 ^b^ ± 1.8	5.73 ^c^ ± 0.2	23.3 ^c^ ± 0.20	70.3 ^c^ ± 5.7
BRS 305 (brown)	90.5	20.11	1.86	0.29	2.76	0.37	13.9 ^c^ ± 0.3	10.8 ^b^ ± 0.2	1.81 ^c^ ± 0.01	36.1 ^a^ ± 0.6	69.9 ^a^ ± 0.7	16.9 ^a^ ± 0.6	8.7 ^d^ ± 0.37	305 ^a^ ± 39.4
1167048 (brown)	98.0	21.60	1.94	0.43	2.90	0.37	12.5 ^d^ ± 0.3	9.5 ^d^ ± 0.1	2.90 ^a^ ± 0.02	14.4 ^c^ ± 0.3	42.8 ^b^ ± 0.6	9.43 ^b^ ± 0.1	9.0 ^d^ ± 0.29	195.3 ^b^ ± 14.3

* Results expressed on a dry basis. TEV: total energy value (kcal), calculated by using Atwater coefficients; CHO (g/100 g), PR: protein content (g/100 g), LI: Lipid content (g/100 g), FIB (g/100 g), ASH (g/100 g), MOI: moisture (g/100 g); RS: resistant starch (g/100 g), TPC: total phenolic content (mg), TAN: total condensed tannins (mg of proanthocyanidins/mL), DPPH IC50: 2,2-diphenyl-1-picryl-hydrazyl-hydrate inhibitory concentration (mg/mL), FRAP TAA: ferric reducing ability power total antioxidant activity (mMTeq/g). Values are means ± standard deviation (*n* = 3). Means with the same letter in the same column do not differ significantly by the Tukey test (*p* > 0.05).

**Table 3 foods-11-03124-t003:** Pasta cooking traits with sorghum flour as 50% of the starch source.

Variety	Optimal Cooking Time (min)	Water Absorption Index (WAI)	Volume Gain (mL)	Soluble Solids Loss (%)
Commercial (white)	14.8	3.30 ^bc^ ± 0.03	9.4 ^a^ ± 0.53	5.30 ^b^ ± 0.70
CMSXS 180 (white)	15.0	3.33 ^bc^ ± 0.18	10.1 ^a^ ± 1.15	5.74 ^b^ ± 0.95
BRS 332 (red)	15.2	3.18 ^c^ ± 0.12	10.3 ^a^ ± 1.15	5.53 ^b^ ± 0.30
BRS 330 (red)	15.1	3.66 ^a^ ± 0.07	10.7 ^a^ ± 0.58	7.97 ^a^ ± 0.78
BRS 305 (brown)	15.3	3.48 ^ab^ ± 0.07	10.3 ^a^ ± 0.58	6.96 ^ab^ ± 0.98
1167048 (brown)	15.3	3.24 ^bc^ ± 0.12	10.0 ^a^ ± 1.00	6.08 ^ab^ ± 0.65

Values are means ± standard deviation (*n* = 3). Means with the same letter in the same column do not differ significantly by the Tukey test (*p* > 0.05).

**Table 4 foods-11-03124-t004:** Sensory acceptance of the six sorghum GFPa.

Trait	Sampling	Commercial (White)	CMSXS 180 (White)	BRS 332 (Red)	BRS 330 (Red)	BRS 305 (Brown)	1167048 (Brown)	*p* Value
Overall quality	General (*n* = 121)	5.45 ^A^ ± 2.00	5.37 ^AB^ ± 1.66	5.31 ^AB^ ± 1.67	5.36 ^AB^ ± 1.59	4.68 ^B^ ± 1.80	5.03 ^AB^ ± 1.78	0.024
	Cl.1 (*n* = 41)	6.88 ^a^	6.63 ^a^	6.68 ^a^	6.63 ^a^	6.34 ^a^	6.34 ^a^	0.341
	Cl.2 (*n* = 80)	4.73 ^Ab^	4.73 ^Ab^	4.61 ^ABb^	4.70 ^Ab^	3.98 ^Bb^	4.36 ^ABb^	0.013
Flavor	General (*n* = 121)	5.15 ^AB^ ± 1.38	5.2 ^AB^ ± 1.63	5.35 ^A^ ± 1.42	4.81 ^B^ ± 1.65	4.25 ^C^ ± 1.07	4.88 ^AB^ ± 1.57	0.000
	Cl.1 (*n* = 41)	6.32 ^a^	6.42 ^a^	6.54 ^a^	5.98 ^a^	5.63 ^a^	6.37 ^a^	0.129
	Cl.2 (*n* = 80)	4.55 ^Ab^	4.58 ^Ab^	4.74 ^Ab^	4.21 ^ABb^	3.58 ^Bb^	4.11 ^ABb^	0.001
Odor	General (*n* = 121)	5.81 ± 1.24	5.68 ± 1.55	5.69 ± 1.03	5.85 ± 1.63	5.55 ± 1.29	5.60 ± 1.35	0.571
	Cl.1 (*n* = 41)	6.49 ^a^	6.24 ^a^	6.49 ^a^	6.15 ^a^	6.27 ^a^	6.39 ^a^	0.783
	Cl.2 (*n* = 80)	5.46 ^b^	5.39 ^b^	5.28 ^b^	5.70 ^a^	5.19 ^b^	5.20 ^b^	0.182
Color	General (*n* = 121)	4.99 ± 1.56	4.68 ± 1.56	4.75 ± 1.47	4.76 ± 1.37	4.67 ± 1.44	4.77 ± 1.63	0.832
	Cl.1 (*n* = 41)	6.15 ^a^	5.78 ^a^	6.20 ^a^	5.93 ^a^	6.10 ^a^	6.02 ^a^	0.896
	Cl.2 (*n* = 80)	4.40 ^b^	4.11 ^b^	4.01 ^b^	4.16 ^b^	3.91 ^b^	4.13 ^b^	0.682

Cl.: Cluster. Means with different capital letters in the same line differ significantly (*p* < 0.05). Means with different small letters in the same column differ significantly (*p* < 0.05).

**Table 5 foods-11-03124-t005:** Ranking descriptive analysis for the six sorghum GFPa.

Descriptive Data	Commercial (White)	CMSXS 180 (White)	BRS 332 (Red)	BRS 330 (Red)	BRS 305 (Brown)	1167048 (Brown)
Brown color	43 ^a^	74 ^ab^	108 ^b^	107 ^b^	169 ^c^	152 ^c^
Sandy surface	60 ^a^	111 ^bc^	106 ^b^	105 ^b^	149 ^c^	121 ^bc^
Whole-grain odor	86 ^a^	107 ^ab^	106 ^ab^	96 ^ab^	134 ^b^	124 ^ab^
Pasta odor	81 ^a^	111 ^ab^	110 ^ab^	96 ^ab^	120 ^ab^	135 ^b^
Sweetness	132 ^b^	115 ^ab^	109 ^ab^	110 ^ab^	84 ^a^	102 ^ab^
Whole-grain flavor	86 ^a^	108 ^ab^	105 ^ab^	102 ^ab^	130 ^b^	122 ^ab^
Hardness	98 ^a^	105 ^a^	90 ^a^	115 ^a^	128 ^a^	117 ^a^
Granulosity	60 ^a^	112 ^bc^	118 ^bc^	98 ^ab^	144 ^c^	120 ^bc^

Sums of values followed by the same letter do not differ by the Friedman test (*p* > 0.05).

## Data Availability

Data is contained within the article and as Appendix A.

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
