# Peer review of "Gluten-Free Sorghum Pasta: Composition and Sensory Evaluation with Different Sorghum Hybrids"

_foods, 2022, doi:10.3390/foods11193124_

Round 1
Reviewer 1 Report
This MS describes some properties of fresh noodles produced from various sorghum cultivars. Although the topic is of interest – in particular due to the fact that sorghum is a sustainable crop - there are a few points where the experimental approach is either not clear or inappropriate, along with several misleading and unclear points that should be clarified.
Methodological issues
It is not clear if all the characterization of phenolic compounds was made on uncooked or cooked pasta. In this context, there are many studies that indicate a significate loss of phenolic compounds after pasta is cooked. Therefore, it is fundamental to perform these determinations also on the cooked samples in order to provide evidence of the real amount of bioactives that is ingested when pasta is eaten. These data should be also important as for defining the capacity of pasta making process in incorporating these bioactive compounds after cooking. If the determinations were performed on raw samples, all the comments on the phenolic properties in the various pasta samples are meaningless. This observation must be taken into account also when the Authors comment the effect of these compounds in terms of the digestion of starch, in particular when it comes to the RS fraction.
The nutritional composition section, made by printout data add some further confusion as for understating the main message obtained from this investigation. The data of nutritional composition may be summarized in a few lines in the 2.2 (pasta development) section under M&M. In the opinion of this Reviewer all the parts that deal with comments on the “nutritional composition” should be removed, as it appears appendicular and poorly related to the purpose of this report.
Introduction
Please reconsider the text, trying to present the sentences with a logical order. The present version is more similar to a cut-and-paste job, in which sentences are combined without any rationale, so that reading this is very difficult. As an example, the sentence on sorghum crop cultivation (line11-117) should be part of the section in which sorghum is described.
There are also some trivial sentences that should be removed. The first part of introduction and the relative box 1 (line 34-40) represent blatant examples of out-of-scope material.
The aim of this investigation it is not clear and should be added at the end of this section
Line 79: raw(??) pasta has many advantages in terms of storage and transport.
Abstract
The last sentence (line 38-39) contradicts what is stated few lines above on the difference among the various pasta samples.
Results
All this section should be reconsidered taking into account the methodological issues discussed at length in the early remarks from this Reviewer (please see above).
Author Response
Dear Reviewer,
Thanks for the essential considerations. Based on "the review report form" evaluations and the two other reviewers, we made many corrections and modifications to improve the manuscript.

Reviewer 2 Report
This study evaluated the development of gluten-free sorghum pasta using six sorghum hybrids. The subject of this study is interesting, and results are novel and promise. The methodologies still can be improved and the sequence of discussion need major revamps. The sequence of discussion should follow the sequence of methodologies in order for the reader to understand easily. The detailed comments are as below:
1. Line 34 – Species name should be written in italic format.
2. Line 35 - Box 1 change to Table 1
3. Line 93 – Box 1 change to Table 1
4. Line 67 – Provide full name for TAN
5. Line 132 – 133 – Provide the brand name of each raw materials used (where necessary)
6. Line 133 – Describe the details mixing process and the instrument used.
7. Line 161 – Falcon tube – should use lower case
8. Line 171 – provide full name for DMAC
9. Line 186 – Provide full name for VG and SSL
10. Line 207 – 212 - Where did the testing take place? What were the sample presentation conditions? Were three-digit codes used? Was a palate cleanser used? Elaborate the details of the hedonic scale used.
11. Why didn’t texture acceptability in sensory analysis?
12. Line 138 – No statistical analysis and SD shown for TEV, CHO, PR, LI, FIB and ASH.
13. Table 1, 2, 3, 4 – Letters in the table should be in superscript format.
14. Line 525 – Incomplete sentence!
Author Response
Dear Reviewer,
Firstly, thanks for the essential considerations. Based on "the review report form" evaluations and the two other reviewers, we made many corrections and modifications to improve the manuscript.
We organized the methodologies in such a way that the sequence matches that of the results and discussion

Reviewer 3 Report
I reviewed the manuscript entitled, Gluten-free sorghum pasta: composition and sensory evaluation with different sorghum hybrids. The study has no novelty. There are many studies dealing with preparation of Gluten-free sorghum pasta. For example, https://doi.org/10.1002/jsfa.9310 .
The similar titles is available at https://doi.org/10.1016/j.jand.2019.08.093 and https://www.jandonline.org/article/S2212-2672(19)31223-7/fulltext#relatedArticles
Author Response
Thanks for the essential considerations. Based on "the review report form" evaluations and the two other reviewers, we made many corrections and modifications to improve the manuscript.

Round 2
Reviewer 1 Report
The Authors have taken due care of the remarks from this Reviewer, at least in those cases where relevant issues were at stake.
Reviewer 3 Report
Authors are now answered the questions raised by me. In my opinion, this version can be accepted for publication in this journal.